# Degradation Characteristics and Remediation Ability of Contaminated Soils by Using *β*-HCH Degrading Bacteria

**DOI:** 10.3390/ijerph20042767

**Published:** 2023-02-04

**Authors:** Qing Chen, Huijun Shi, Yanpeng Liang, Litang Qin, Honghu Zeng, Xiaohong Song

**Affiliations:** 1College of Environmental Science and Engineering, Guilin University of Technology, Guilin 541004, China; 2Guangxi Key Laboratory of Environmental Pollution Control Theory and Technology, Guilin University of Technology, Guilin 541004, China; 3Collaborative Innovation Center for Water Pollution Control and Water Security in Karst Region, Guilin University of Technology, Guilin 541004, China

**Keywords:** *β*-HCH, microbial degradation, soil remediation, microbial community

## Abstract

Three degradation strains that can utilize *β*-Hexachlorocyclohexanes (*β*-HCH) as the sole carbon source were isolated from the soil substrate of constructed wetland under long-term *β*-HCH stress, and they were named A1, J1, and M1. Strains A1 and M1 were identified as *Ochrobactrum* sp. and strain J1 was identified as *Microbacterium oxydans* sp. by 16S rRNA gene sequence analysis. The optimum conditions for degradation with these three strains, A1, J1, and M1, were pH = 7, 30 °C, and 5% inoculum amount, and the degradation rates of 50 μg/L *β*-HCH under these conditions were 58.33%, 51.96%, and 50.28%, respectively. Degradation characteristics experiments showed that root exudates could increase the degradation effects of A1 and M1 on *β*-HCH by 6.95% and 5.82%, respectively. In addition, the degradation bacteria A1 and J1 mixed in a ratio of 1:1 had the highest degradation rate of *β*-HCH, which was 69.57%. An experiment on simulated soil remediation showed that the compound bacteria AJ had the best effect on promoting the degradation of *β*-HCH in soil within 98 d, and the degradation rate of *β*-HCH in soil without root exudates was 60.22%, whereas it reached 75.02% in the presence of root exudates. The addition of degradation bacteria or degradation bacteria-root exudates during soil remediation led to dramatic changes in the community structure of the soil microorganisms, as well as a significant increase in the proportion of aerobic and Gram-negative bacterial groups. This study can enrich the resources of *β*-HCH degrading strains and provided a theoretical basis for the on-site engineering treatment of *β*-HCH contamination.

## 1. Introduction

Hexachlorocyclohexanes (HCHs) are a class of organochlorine pesticides (OCPs) and persistent organic pollutants (POPs) that have attracted worldwide attention [1]. Since the 1970s, the usage of HCHs have been banned by many countries all over the world. However, due to the long half-life and high stability of HCHs, they can still be detected globally and have certain environmental risks [2,3,4]. Among all HCH isomers, the *β*-isomer (*β*-HCH) is the most stable, with a half-life of up to 30 years in the natural environment [5]. It also has a very low water solubility, is biotoxic, is a high-risk endocrine disruptor, and is a carcinogenic, teratogenic pollutant that easily bioaccumulates [6,7,8]. Compared with other isomers, *β*-HCH has lower solubility and a higher vapor pressure. Therefore, *β*-HCH more easily remains in sediment, soil substrate, and the atmosphere and has strong mobility [9]. The degradation of HCHs in nature is mainly dependent on the microorganisms in the environment [10]. In recent years, chemical oxidation combined with microbial degradation of HCHs has been an efficient technology, but the technical conditions are complex with relatively high costs [11,12]. The combination of plant and microbial methods is economically beneficial [13]; plants enrich the community structure of soil microorganisms and root secretions can facilitate the degradation of HCHs by microorganisms [14]. Currently, many microorganisms with a good degradation ability for *β*-HCH have been reported, such as *Sphingobium* sp. [15], *Sphingomonas* sp. [16], *Streptomyces* sp. [17], *Ochrobactrum* sp., and *Pseudomonas* sp. [18]. In particular, *Sphingobium* sp. has abundant genes related to the degradation of *β*-HCH [19]. However, most of these studies are limited to high-concentration treatments or beaker experiments, and few studies have been applied to soil remediation.

In our previous study, we found that the entry of *β*-HCH into a constructed wetland (CW) significantly altered the substrate microbial community and that the degradation rate of *β*-HCH in the substrate increased as the microbial response became greater [20]. Microbial communities in CWs are usually more complicated and tend to show greater adaptability to pollutants than those in soil [21]. Microbial degradation is an economical method with minimal environmental impact [22], so there is a lot of promise in screening the environment for well-adapted strains that can degrade well at low concentrations of *β*-HCH. The goals of this study are to find bacteria that have good degradation abilities for *β*-HCH and can be used in the treatment of soil *β*-HCH contamination, even under low-*β*-HCH-concentration conditions and provide a theoretical basis for the on-site engineering treatment of *β*-HCH contamination.

## 2. Materials and Methods

### 2.1. Experimental Materials

The soil used for the enrichment of degrading bacteria was taken from a model constructed wetland at the water station of Guilin University of Technology, whereas the contaminated soil was taken from the Huixian wetland in Guilin, Guangxi, China.

Inorganic culture medium: K_2_HPO_4_ 1 g, KH_2_PO_4_ 1 g, NH_4_NO_3_ 1 g, MgSO_4_·7H_2_O 0.5 g, CaCl_2_ 0.02 g, FeCl_2_ 0.01 g, reverse osmosis (RO)-water 1000 mL, and pH = 7.

Luria-Bertaniculture (LB) culture medium: bacteriological peptone 10 g, yeast 5 g, NaCl 10 g, RO-water 1000 mL, and pH = 7. The prepared culture medium was sterilized at 121 °C for 20 min in an autoclave, and 15 g of agar powder was added to the medium to sterilize it in the same way when the solid culture medium was prepared. The simulated root exudates were prepared according to the method reported in the literature [23], and 100 μg/L of *β*-HCH stock solution was prepared with acetone as the solvent for use.

Main reagents: *β*-HCH standard solution (100 μg/L, purity > 99.0%) was purchased from the Ministry of Ecology, Agriculture and Environment, China. Acetone and methanol were purchased from TEDIA Company, Ohio, United States of America. Ethyl acetate and methylene chloride were purchased from Jingchun Biochemical Technology Co., Ltd., Shanghai, China. All the used chemicals were chromatographically pure.

### 2.2. Enrichment, Screening, and Isolation of Degrading Bacteria

The CW model (dimensions: 1.0 × 0.5 × 0.6 (unit: m), number: 8, with a uniform laying down of cobblestone with a height of 10 cm and a particle size of about 2.0–3.0 cm at the bottom of the vessel and a 20 cm-thick layer of soil laid on top of the cobbles planted with *Acorus calamus*.) with the lowest background value of *β*-HCH was selected to collect 10 g of fresh plant inter-root soil in 100 mL of inorganic medium with *β*-HCH (concentration controlled at 1 mg/L) as the only carbon source, and then this was incubated for 7 d at 30 °C in a constant-temperature shaker at 160 r/min. Then, the supernatant suspension was taken and cultured for 7 d under the same conditions, and enrichment was repeated 5 times. The last culture was diluted in a gradient of 10^−3^ to 10^−5^ and spread on inorganic plate medium containing 1 mg *β*-HCH. After incubation in a constant-temperature incubator at 30 °C for 7 d, single colonies with rapid growth and obvious colonies were selected and then repeatedly isolated by scratching on an LB plate medium until pure strains were obtained. All these operations were performed in a vertical-flow clean bench. The pure strains with good growth and stable passage were selected and inoculated into inorganic medium (pH = 7) containing 50 μg/L of *β*-HCH to verify the degradation ability. Finally, the three strains with the best degradation effect were selected and named as A1, J1, and M1 for subsequent studies.

### 2.3. Strain Identification

The strains were identified by 16S rDNA sequencing, which was entrusted to Sangon Biotech Co., Ltd., Shanghai, China. The obtained sequencing results were matched on NCBI (www.ncbi.nlm.nih.gov, accessed on 15 November 2021.) using the blast, and then the sequence of the strain obtained by sequencing and the 16S rDNA sequence of the known strain with the highest homology were used to construct a phylogenetic tree by MEGA7.

### 2.4. Bacterial Growth Curve and Preparation of Bacterial Suspension

Bacterial growth: The three purified strains in Section 2.2 were inoculated in 100 mL of LB culture medium and incubated at 30 °C in a constant-temperature shaker at 160 r/min. The OD_600_ of the bacterial solution was measured every 6 h and the growth curve was plotted.

Bacterial suspension: The three activated strains were incubated in 100 mL of LB culture medium at 30 °C on a constant-temperature shaker at 160 r/min until the logarithmic growth phase. Then, the bacterial solution was centrifuged at 7000 r/min for 5 min. The supernatant was discarded, the organisms were collected after rinsing two times with inorganic medium in a clean bench, and finally the organisms were suspended in PBS buffer (pH = 7) to obtain a bacterial suspension.

### 2.5. Degradability and Degradation Characteristics of the Strains

#### 2.5.1. Degradability of Strains

The bacterial suspensions were inoculated into inorganic medium (pH = 7) containing 50 μg/L β-HCH at 1% volume ratio for different degradation times (1, 3, 5, 7 and 14 d), and the uninoculated group were used as blank controls, with three parallel treatments at 30 °C and 160 r/min.

#### 2.5.2. Degradation Characteristics of the Strains

The bacterial suspensions were inoculated into 100 mL of inorganic medium (pH = 7) containing 50 μg/L of *β*-HCH at a volume ratio of 1% set up with pH gradients (pH = 4.0, 6.0, 7.0, 8.0, 10.0), temperature gradients (20, 25, 30, 35, 40 °C), and *β*-HCH concentration gradients (10, 20, 40, 50 μg/L) to determine the degradation of *β*-HCH by the strains. Each treatment was incubated at 30 °C and 160 r/min for 7 d. The growth and *β*-HCH residues were measured on the last day.

Four groups were set up in the external nutrition experiment: 100 mL of inorganic medium (pH = 7) containing 50 μg/L of *β*-HCH containing 1% bacterial suspension was used for the control group (CK). Three groups of external nutrient fractions were added to the control group, with 0.5 g of root exudates(R-EX), methanol, and yeast powder used, respectively. The samples without the inoculated bacterial solution were used as blank controls (K), and three parallels were set up for each treatment. The incubation conditions were 30 °C and 160 r/min for 7 d. Samples were taken every 24 h to determine the growth of the strains and on the last day to determine the *β*-HCH residue.

### 2.6. Construction of Complex Bacteria

The prepared suspensions of each degrading bacterium were mixed in pairs according to a volume ratio of 1:1 to produce 1 mL of a mixture, which was named AJ, AM and JM according to strains A, J and M. The three strains of bacteria were mixed together at a volume ratio of 1:1:1 to prepare a 1 mL mixed bacterial preparation, which was recorded as AJM. The prepared mixed bacterial agent was inoculated into 100 mL of inorganic medium containing 50 μg/L of *β*-HCH at a ratio of 1%, and then cultured under shaking at 30 °C and 160 r/min for 7 d in a shaker. The residual amount of *β*-HCH was finally measured on the last day.

### 2.7. Simulation of Contaminated Soil Remediation Trials

In this operation, 500 g of dry CW substrate soil was placed in a 1 L beaker containing 10 μg of *β*-HCH and an appropriate amount of ultrapure water. Then, each of the three strains was inoculated with 5% inoculum. A separate control experiment was set up under the same conditions, except that 1 mL of rhizosphere exudates was added to the beaker and seven small models were made for each bacterium group. Soil samples were collected at 1, 14, 28, 42, 56, 70, 84, and 98 d to determine the residues of *β*-HCH.

Soils from the *β*-HCH degradation experiment were collected for microbial sequencing. Since a significant amount of *β*-HCH degradation began in the soil at 56 d and there was a significant decrease in *β*-HCH degradation at 84 d, soils from groups A, A_R, J, J_R, AJ, and AJ_R at day 56 of the run and from groups A, A_R, J, J_R, AJ, and AJ_R at day 84 were collected for the study. Soil samples from day 56 were recorded as A4, A4_R, J4, J4_R, AJ4, and AJ4_R, respectively, and those from 84 d were recorded as A6, A6_R, J6, J6_R, AJ6, and AJ6_R, respectively, and compared with the K and K_R groups at 7 d as CK1 and CK1_R. All samples were stored in a refrigerator at −80 °C and sent for sequencing at the same time.

### 2.8. Sample Pre-Treatment and Detection

Sample pre-treatment: The treatment was carried out according to that reported in the literature [20], except that the bacterial solution was first centrifuged at 7000 r/min for 5 min in a freezing centrifuge before being treated as an aqueous sample.

GC/ECD chromatographic conditions [24]: The content of *β*-HCH in the sample was subjected to the GC-ECD analysis after solid phase extraction. The sample was injected by an autosampler with a splitting ratio of 10:1 and an injection volume of 0.8 μL. The temperature of the injection port and ECD detector were set at 250 °C and 320 °C, respectively. High-purity nitrogen was used as the carrier gas; each sample was measured for 25 min and the peak time for *β*-HCH was 18.4 min.

## 3. Results and Discussion

### 3.1. Bacterial Isolation, Screening and Identification

The phylogenetic tree was constructed on the basis of blast results as shown in Figure 1. Three strains that could use *β*-HCH as the sole carbon source and had the ability to degrade *β*-HCH were isolated by enrichment technology, and they were named as A1, J1, and M1. According to the results of the 16S rDNA gene sequence analysis, strains A1 and M1 showed the highest degree of similarity (99%) with *Ochrobactrum*. They were identified as *Ochrobactrum* sp. strain A1 (accession number OP010035) and *Ochrobactrum* sp. strain M1 (accession number OP010036), respectively. Strain J1 showed the highest degree of similarity (99%) with *Microbacterium*. It was identified as *Microbacterium* sp. strain J1 (accession number OP050394). The morphological characteristics of the colonies of degraded strains A1, M1, and J1 are shown in Appendix A whereas their morphophysiological and physiochemical properties are listed in Appendix A. By combining the morphological characteristics, physiological and biochemical properties, and 16S rDNA sequencing results, strains A1 and M1 were finally identified as *Ochrobactrum* sp. while strain J1 was detected as *Microbacterium oxydans* sp.

### 3.2. Growth Curve and Degradation Ability of Strains

The growth of each strain in the inorganic medium showed a linear increase and then gradually leveled off. All three strains showed significant logarithmic growth during the incubation process, and the logarithmic growth period was from 3 d to 7 d. The OD_600_ value in the inorganic medium of A1 showed a higher growth rate with time (Figure 2a). This indicated that strain A1 grew faster and had more biomass in a short time when only *β*-HCH was available as a carbon source. Comparing the degradation rates of the three strains within 14 d, it can be seen in Figure 2b that strain A1 has a higher ability to utilize *β*-HCH, eventually degrading 58.59% of *β*-HCH within 14 d, followed by strains J1 (47.16%) and M1 (46.65%). *β*-HCH is difficult to biodegrade because of its unique structural stability, and some bacteria, despite their ability to degrade *β*-HCH, generally do not perform well under low concentrations of *β*-HCH treatment. Many bacteria have good degradation ability for other isomers of HCHs, but almost none of them can degrade *β*-HCH [8]. In previous reports, *Sphingomonas* sp. NM05 [15] could degrade about 60% of *β*-HCH (40–60 ug/mL) and *Streptomyces* sp. M7 [17] degraded 78% of *β*-HCH at 1.66 mg/L. Compared with these reports, the bacteria reported in this paper also have better degradation ability at lower *β*-HCH concentration, and may have a better possibility to be applied in engineering practice.

Overall, the degradation capacity of the strains was positively correlated with their biomass in the first 7 d. In addition, the OD_600_ values of the groups stabilized after 7 d of the bacterial solution reaction, while the *β*-HCH degradation rate did not differ significantly between the 7th and 14th day (*p* > 0.05). The changes in OD_600_ values showed that, although there was no significant decrease in the number of organisms in the bacterial solution from 7 d to 14 d, the utilization of *β*-HCH by the degrading bacteria had been significantly reduced, and strains A1, J1, and M1 only degraded 1.57%, 0.42%, and 1.67% of *β*-HCH from day 7 to 14, respectively. This may be due to the metabolic inhibition of the bacterium caused by the accumulation of metabolic wastes in the media system. A study has shown that changes in pH, accumulation of metabolic waste, and temperature could affect the removal of contaminants by bacteria [25]. In this study, it was found that the pH of each medium decreased after 7 d of incubation, with an average of pH = 5.42 ± 0.15, which may be an important factor affecting the significant decrease in the degradation of *β*-HCH by the strain after 7 d.

### 3.3. Degradation Characteristics of the Strains

Appendix A shows that the increase in inoculum had no significant effect on the final growth of the strains (*p* > 0.05). The highest *β*-HCH degradation rate was observed in all groups when the inoculum was 5%, but when the inoculum increased from 1% to 5%, the contribution of A1, J1, and M1 to degrading *β*-HCH was only increased by 2.71%, 3.55%, and 3.9%, respectively (Figure 3a), which was not significant (*p* > 0.05). In the pH-sensitive experiments on the growth environment of the strains, we found that the growth of the strains was inhibited to different degrees under both acidic and alkaline conditions, while the growth status of each strain was better under neutral or alkaline pH conditions (Appendix A). Furthermore, all three strains had a good degradation effect in the pH = 6–8 range. Extreme acidic and alkaline environments could seriously inhibit the degradation of *β*-HCH by the strains, and each strain showed the greatest degradation ability of *β*-HCH under neutral pH conditions. The degradation rates of A1, J1, and M1 for *β*-HCH were 56.11%, 48.69%, and 47.31%, respectively (Figure 3b). The results for ambient temperature showed that a constant temperature of 30 °C was the best degradation temperature (Figure 3c) and the best growth temperature (Appendix A) for the three strains, resulting in degradation rates for A1, J1, and M1 for *β*-HCH were 56.54%, 49.86%, and 47.37%, respectively. Neither high nor low temperatures could be conducive to the growth of the strain and the degradation of *β*-HCH. Previously, it was shown that lower temperatures would weaken the metabolism of most microorganisms, while higher temperatures would greatly inhibit the growth and enzymatic activity of the microorganisms [26,27]; coincidentally, the degradation of *β*-HCH is associated with a number of biological enzymes [28].

The initial concentration of substrate *β*-HCH had no significant effect on the final growth of the three strains (Appendix A), but the degradation rate of *β*-HCH by the three strains showed a trend of increasing and then decreasing with further increases in the initial concentration of *β*-HCH (Figure 3d), and each strain showed the highest degradation efficiency of *β*-HCH at the concentration of 50 μg/L. The degradation efficiency of *β*-HCH by strains A1, J1, and M1 was 57.01%, 47.29%, and 46.14%, respectively. Compared to the substrate concentration of 50 μg/L, the ability of the strains to degrade *β*-HCH was significantly reduced when the concentration of *β*-HCH was lower. This phenomenon may be due to the fact that *β*-HCH was the only carbon source and the underutilization by the strains of *β*-HCH in a short period of time; accompanied by changes in the metabolic environment (such as pH reduction and production of harmful by-products, etc.) [29], these factors weakened the degradation of *β*-HCH by the strains. When the concentration of *β*-HCH was high, the reduction in degradation rate may be related to the physiological toxicity of *β*-HCH [12].

Under the treatments with additional carbon sources, methanol increased the growth of strains A1, J1, and M1 by an average of 2.67%. The root exudates increased the growth of strains A1, J1, and M1 by an average of 11.82%. The yeast powder significantly increased the growth of strains A1, J1, and M1, with an average of 32.55% (Appendix A). The degradation of *β*-HCH was not always proportional to growth, although the growth of the strains was enhanced to varying degrees by the addition of other carbon sources (Figure 3e). The results showed that the addition of root exudates increased the degradation of *β*-HCH by 6.95% and 5.82% for strains A1 and M1, respectively, but led to a 7.71% decrease in the degradation of *β*-HCH by strain J1, which may be due to the simulated formulated root exudates containing a large amount of nutrients such as aromatic hydrocarbons. The utilization of carbon sources by strain J1 was more inclined towards aromatic hydrocarbons, which led to the selectivity of strain J1 for *β*-HCH. In addition, the role of methanol is to act as an auxiliary carbon source to increase the number of degrading bacteria, thus promoting the degradation of the target pollutant by the degrading bacteria [30]. However, the addition of methanol had less auxiliary effect on the degradation of *β*-HCH by the strains in this experiment. The presence of the yeast powder improved the medium adsorption rate of *β*-HCH, with about 57.47% of *β*-HCH being adsorbed rather than being degraded. The degradation rate of *β*-HCH by strains A1 and J1 was reduced by 19.65% and 6.88%, respectively, in the culture with the addition of yeast powder, while the reduction rate of *β*-HCH by strain M1 was as high as 34.41% at 7 d. This was most likely due to the adsorption of yeast powder [31], which prevented *β*-HCH from being utilized by bacteria in the aqueous phase.

### 3.4. Study of Optimal Mix of Degradation Bacteria

The degradation effects of the compound bacteria on *β*-HCH are shown in Figure 4. The composite bacterial agent AJ of strains A1 and J1 was the most effective and degraded 69.57% of the *β*-HCH at 7 d. However, the composite bacterial agents AM, JM, and AJM only showed degradation efficiencies of 36.85%, 33.05%, and 24.84%, respectively. It could be seen that the degradation of *β*-HCH was improved by only the mixture of A1 and J1 compared with that of the single bacteria, while the degradation of *β*-HCH by the other mixtures was generally reduced. Jing et al. [32] screened three strains of ammonia-oxidizing bacteria from sludge and found that these three strains enhanced the conversion of ammonia nitrogen from 23.3 mg/L/d to 47.09 mg/L/d with a significant effect after compounding by experiments. Virgilio et al. [33] found that certain mixtures of bacteria could significantly enhance the degradation of some pesticides. However, it was also noted in their study that mixed bacteria showed unsatisfactory degradation of 2,4-D-carbofuran and diazinon, but single strains showed better degradation of these pesticides. A study also indicated that this observation can be related to synergistic and antagonistic interactions between microorganisms [34]. We deduced that the antagonism between degrading bacteria A1 and J1 in this study was very low, and there might also be some synergistic effect. In addition, a strong competition between A1 and J1 bacteria and M1 bacteria was obviously unfavorable for the degradation of *β*-HCH.

### 3.5. Remediation Study of Simulated β-HCH Contaminated Soil

The variation of *β*-HCH content in soil under different degrading bacteria treatments is shown in Figure 5a, which demonstrated a trend of increasing and then decreasing with the increase in treatment time. The *β*-HCH content in the soil increased continuously from day 1 to day 7 of the experiment and reached the maximum in each group at 7 d. This phenomenon was caused by the gradual adsorption of *β*-HCH from the aqueous solution into the system by the soil, and this process was also accompanied by a mutual exchange between the water and the soil until a dynamic equilibrium was reached [20]. Many pesticides, especially hydrophobic pesticides, are heavily adsorbed into the soil substrate after entering a stream or wetland system, which is related to the n-octanol/water partition coefficient (*Kow*) value of the pesticide [35]. A higher *Kow* value indicates a higher sorption coefficient in the soil or sediment [36,37]. The subsequent decrease in *β*-HCH content was due to the degradation by microorganisms in the soil, and the slope of the curve in the figure indicates the rate of *β*-HCH degradation. The changes in the degradation rate of *β*-HCH in soil are shown in Figure 5b. When the experiment was conducted for 7 d, the *β*-HCH content in soil began to decrease; in the 7th to 14th d, the reduction rate of *β*-HCH content reduction was slower, which might be related to the fact that the bacterial agent at this stage was in the adaptation period. *β*-HCH is also a kind of hard-to-degrade organic substance with a certain biological toxicity; its structural stability determines that it is difficult to degrade in a short-term period [38]. The degradation rate of *β*-HCH started to increase gradually from the 14th d. Finally, the levels of *β*-HCH in groups A, J, AJ, and CK were 11.78, 10.85, 10.32, and 19.61 μg/kg, and the degradation rates of *β*-HCH in each group were 54.89%, 58.51%, 60.22%, and 24.94%, respectively.

It was noted that group K, without any treatment, could also degrade nearly one-fourth of the *β*-HCH, which indicated that some primitive microorganisms in the soil already had the ability to degrade *β*-HCH and could adapt to its toxic effects. Soil-indigenous microorganisms can adjust and adapt to the disturbed environment and have the potential to progressively metabolize foreign contaminants [39]. In addition, the experimental conditions were conducted at a constant temperature of 30 °C, which attenuated the effect of ambient temperature on soil microorganisms. The effect of temperature on soil microorganisms is crucial, and a suitable temperature can greatly enhance the metabolic capacity of soil microorganisms, thus promoting the decomposition and transformation of pollutants [40]. Figure 5c shows the changes in *β*-HCH content in the soil with time. The changes in the *β*-HCH degradation rate with time when different strains of bacteria were treated with the addition of root exudates is shown in Figure 5d. The content of *β*-HCH in the soil of each group decreased more rapidly after 14 d, after the addition of root exudates, and there was a significant reduction in the final content. The addition of root exudates increased the degradation rate of *β*-HCH significantly in the soil under different treatments, and the degradation rate of *β*-HCH in the groups A_R, J_R, AJ_R and K_R finally increased by 10.14%, 11.56%, 14.81%, and 11.96%, respectively, compared with the experimental group without the addition of root exudates, with the most significant increase in group AJ. Therefore, it can be inferred that the injected strains A1 and J1 and the compound bacterial agent AJ may not have antagonistic effects with soil-indigenous microorganisms, and the coupling with root exudates was more obvious and had the potential for engineering applications.

### 3.6. Characteristics of Microbial Community Response

#### 3.6.1. Alterations in Abundance and Diversity of Microbial Community

The PCA (Principal Component Analysis) plots and clustering trees of soil microorganisms based on OTUs under different treatment conditions are shown in Figure 6. As seen, PC1 and PC2 explained 64.9% of the sample characteristics with high confidence in the data. The groups CK1 and CK1_R were distant from each of the other groups that were injected with degrading bacteria and root exudates, concentrated in the upper left corner of the coordinate system, and it can be visualized that the injection of biofortifiers and root exudates caused significant differences in the soil microbial community compared to the original soil (*p* < 0.05). In addition, when the same degradation bacteria were injected in the system, the sample distance between the unadded root exudates and the injected root exudates was greater. The clustering tree diagram shows the similarity between each sample more visually, and the length of the tree branches represents the distance between samples; the more similar samples are closer together. There was high similarity between A6, J6, and AJ6, and the similarity between A4, J4, and AJ4 was minor, which also indicated that the added bacteria have a greater impact on the diversity of soil microorganisms in the short term, but the soil microbial diversity tends to converge eventually.

Appendix A demonstrates the changes in soil microbial Alpha diversity indices under simulated soil *β*-HCH contamination remediation under the same treatment conditions, and it can be seen that the Simpson index values for each group were small, indicating an extremely rich microbial community diversity. Compared to the CK1 group, the OUTs, Shannon, Chao1, and Ace indices of each group continued to increase during the degradation of *β*-HCH when degrading bacteria were added to the soil. In contrast, the OUTs, Shannon, Chao1, and Ace indices of each group showed an overall increase at the beginning of the experiment compared to the CK1_R group at the end of the experiment, after the soil was injected with both degrading bacteria and root exudates. This suggested that the addition of degrading bacteria can stimulate changes in the background soil microbial community, resulting in a richer microbial community structure, which was also as one of the reasons for the higher degradation rate of *β*-HCH after the addition of degradation bacteria. The lower microbial abundance at the end of the degradation of *β*-HCH in the soil could be due to the degradation bacteria and root exudates gradually leading to a more suitable microbial community structure for *β*-HCH degradation in the soil.

#### 3.6.2. Phylum-Level Characteristics of Microbial Community Structure

The relative abundance of microorganisms in the soil under different treatment conditions in terms of the species in each phylum are shown in Figure 7, with significant differences between each group (*p* < 0.05). Similar microbial fractions of the CK1 and CK1_R groups were observed and mainly contained *Proteobacteria*, *Acidobacteria*, *Firmicutes*, *Chloroflexi*, and *Verrucomicrobia*. These phyla accounted for a total of 86.09% and 82.53% of groups CK1 and CK1_R, respectively. It was found that the relative abundance of *Firmicutes* decreased significantly (*p* < 0.01) after the addition of degradation bacteria or root exudates, and the proportion of them in all groups was on average less than 1%, while the proportion of *Proteobacteria* increased significantly (*p* < 0.05) in all groups, with an average proportion of 42.32%. A notable point is that the relative abundances of *Acidobacteria*, *Bacteroidetes*, *Chloroflexi*, *Nitrospirae*, and *unclassified_Bacteria* increased by 436.73%, 405.54%, 387.04%, 1293.31%, and 391.36%, respectively, after addition of degradation bacteria, compared to the group control. This was 527.64%, 391.44%, 332.22%, 731.73%, and 307.39% higher than the group control after root exudates were added. A study found that *Proteobacteria* and *Acidobacteria* dominated in soil contaminated with HCHs, and the abundance of *Proteobacteria* was significantly increased with the degradation of HCHs [41]. In addition, *Bacteroidetes* was shown to degrade OCPs such as Triclosan [42]. Xu et al. [43] used *Bacteroidetes* to study its degradation effect towards typical OCPs such as Triclosan in pure culture. They found that the Triclosan and other pollutants can be degraded by *Bacteroidetes*. Dehalococcodia is a group of *Chloroflexi* which is known to degrade organohalides. Currently, two genera of this bacterium have been found to be dehalogenating; they are *Dehalococcoides* [44] and *Dehalogenomonas* [45]. In particular, *Dehalococcoides* is among the biomarkers often used for chlorinated solvent contamination, and their detectability in the field is often used as a basic tool to assess the natural attenuation of organochlorine-contaminated sites or the efficiency of bioremediation projects [46,47]. *Nitrospirae* is a group of bacteria that contributes to nitrogen removal from soil and water bodies, and some studies indicated that this group of bacteria have excellent tolerance to salt [48], heavy metals, and some antibiotics [49]. In conjunction with the degradation process and microbial community changes of *β*-HCH in soil, the addition of degradation bacteria or the addition of degradation bacteria and root exudates could significantly change the composition of the soil microbial community at the phylum level and favor *β*-HCH degradation.

#### 3.6.3. Genus-Level Characteristics of Microbial Community Structure

To further clarify the effect of degradation bacteria and the root exudates on the microbial community in the soil, the groups of soil microorganisms were classified at the genus level and the proportions are shown in Figure 8. The differences between the genera of soil microorganisms in each group were highly significant (*p* < 0.01), and the classifications of the soil microorganisms in the groups CK1 and CK1_R were relatively similar at the genus level, but the proportion of each genus varied greatly. Only the relative abundance of *unclassified_Betaproteobacteria*, *unclassified_Anaerolineaceae*, and *unclassified_Deltaproteobacteria* increased with the addition of degradation bacteria and root exudates, while the proportion of all other genera significantly decreased (*p* < 0.05). Among them, the on with the most significant reduction in the proportion was *Clostridium sensu_stricto*, which had proportions of 12.95% and 18.81% in the groups CK1 and CK1_R, respectively, but in the biological augmentation, groups A4, J4, A4_R, J4_R, AJ4, AJ4_R, A6, J6, A6_R, J6_R, AJ6 and AJ6_R contained less than 0.4%. In addition, *unclassified_Pseudomonadaceae* and *unclassified_Oxalobacteraceae* had an average 10-fold lower proportion in the biofortified group. These results indicated that the addition of degradation bacteria, as well as the addition of root exudates, can greatly affect the microbial community structure in the original soil, and this result may be irreversible in the short term.

### 3.7. BugBase Analysis of Soil Microorganisms

The proportion of the oxygen-requiring and Gram-negative bacteria in the soil microorganisms in each group are shown in Figure 9. The addition of degradation bacteria and root exudates favored the development of microorganisms in the direction of aerobic bacteria during the degradation of *β*-HCH in soil, and this was accompanied by a small elevation in the anaerobic flora and a significant decrease in the facultative aerobic flora. In addition, this significantly increased the proportion of Gram-negative flora in the soil microorganisms. There was no significant difference between Group A and B. Compared to the mean relative abundance of Gram-negative bacteria in the original soil, the mean relative abundance of Gram-negative bacteria increased to 98.12% after the biological augmentation was used. These results indicated that dosing of degradation bacteria and addition of root exudates led to a significant increase in the number of Gram-negative bacteria in the soil. Combined with the degradation of *β*-HCH in the soil after biological augmentation, it can be inferred that the degradation bacteria, aerobic conditions, and Gram-negative bacteria in this study may be the key factors in the degradation of *β*-HCH in the soil.

## 4. Conclusions

Three strains, A1, J1 and M1, with good degradation ability for *β*-HCH were isolated from the plant rhizosphere soil of the constructed wetland. The strains A1 and M1 were finally identified as *Ochrobactrum* sp. and strain J1 was identified as *Microbacterium oxydans* sp. by combining their morphological characteristics, physiological and biochemical properties, and 16S rDNA sequencing results. The three strains exhibited optimal degradation of *β*-HCH at pH = 7, temperature 30 °C, inoculation of 5%, and initial *β*-HCH concentration of 50 μg/L. The addition of simulated root exudates can improve the degradation capacity of strains A1 and M1 to *β*-HCH. The degradation capacity of degradation bacteria A1 and J1 for *β*-HCH in a 1:1 compound was the most effective compared to those of others compound bacterial agents.

At a constant temperature of 30 °C, the *β*-HCH content in the soil was reduced by 24.94% due to the degradation of soil native microorganisms. The addition of degradation bacteria significantly increased the degradation of *β*-HCH by soil microorganisms, and, after 98 d of treatment, 54.89%, 58.51%, and 60.22% of *β*-HCH was degraded in the A, J, and AJ groups, respectively. The addition of root exudates further increased the degradation rate of *β*-HCH in the soils of all groups, with 10.16%, 11.57%, and 14.81% increases in groups A, J, and AJ, respectively.

The addition of degradation bacteria led to a change in the soil background microbial community. The initial microbial community structure was more abundant, while the degradation bacteria and root exudates could gradually lead to a soil microbial community structure more suitable for *β*-HCH degradation. The addition of degradation bacteria caused a large impact on the diversity of soil microorganisms in the short term, but the soil microbial diversity still converged at the end of the experiment. Compared with the blank control, the treatment with degradation bacteria resulted in a significant increase in the relative abundance of soil microorganisms at the phylum taxonomic level in *Acidobacteria*, *Bacteroidetes*, *Chloroflexi*, *Nitrospirae*, and *unclassified_Bacteria* in the degradation of soil *β*-HCH. The addition of degradation bacteria favored the development of soil microorganisms toward aerobic flora, increased the proportion of anaerobic flora to a small extent, and decreased the proportion of facultative aerobic flora. The addition of degradation bacteria and root exudates increased the proportion of Gram-negative flora in the soil microorganisms significantly.

## Figures and Tables

**Figure 1 ijerph-20-02767-f001:**
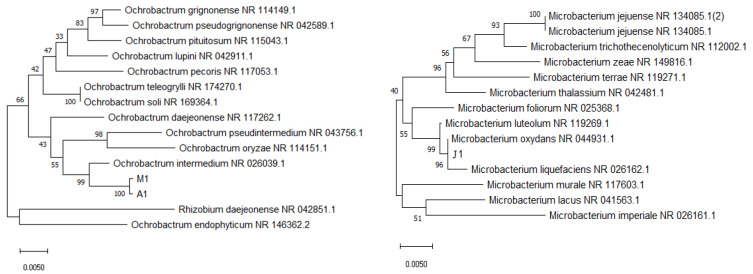
Neighbor-joining phylogenetic tree of the representative bacterial strains A1, J1, and M1, and their related species based on the 16S rRNA gene sequences.

**Figure 2 ijerph-20-02767-f002:**
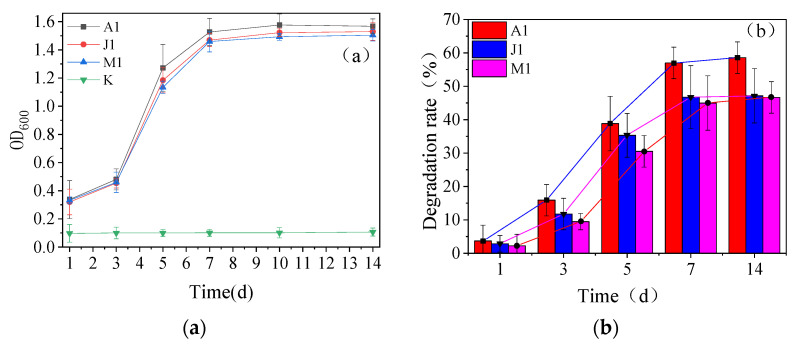
Effects of degradation time on strain growth (**a**) and *β*-HCH degradation (**b**).

**Figure 3 ijerph-20-02767-f003:**
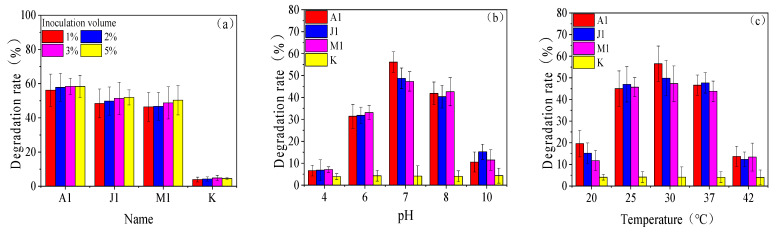
Degradation performance of *β*-HCH by strains at different levels of inoculum (**a**), pH (**b**), temperature (**c**), substrate concentration (**d**), and nutrients (**e**).

**Figure 4 ijerph-20-02767-f004:**
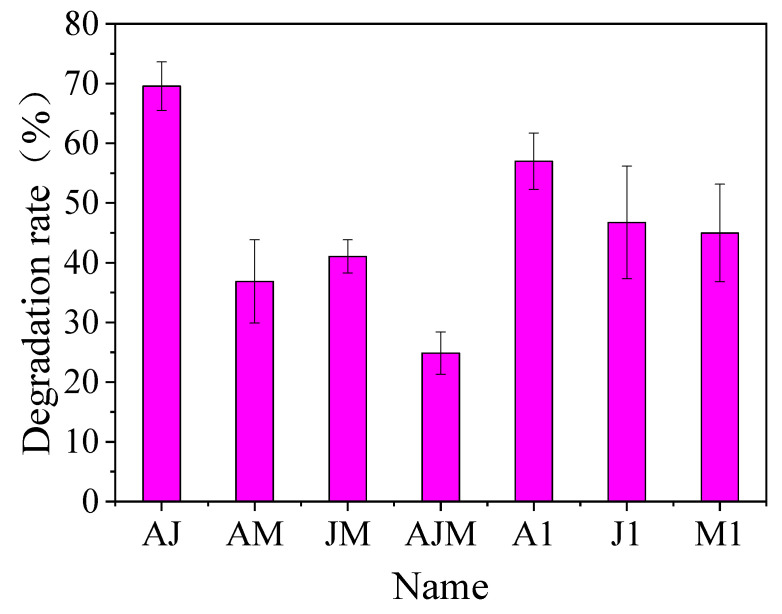
Degradation ability of compound bacterial agents to *β*-HCH (AJ indicates a mixture composed of strains A1 and J1 in equal proportions, and the same for AM, JM, and AJM).

**Figure 5 ijerph-20-02767-f005:**
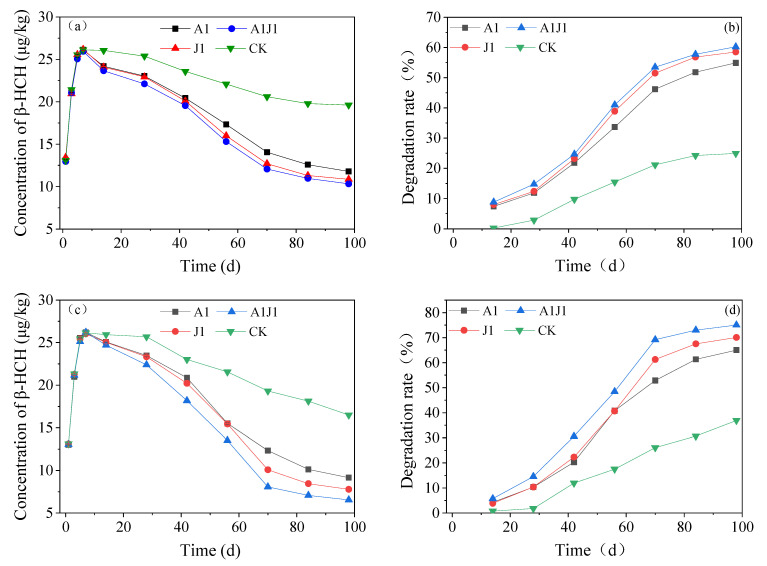
Plots of *β*-HCH content (**a**) and degradation rate (**b**) in soil with conventional treatment and *β*-HCH content (**c**) and degradation rate (**d**) in soil after addition of root secretions over time.

**Figure 6 ijerph-20-02767-f006:**
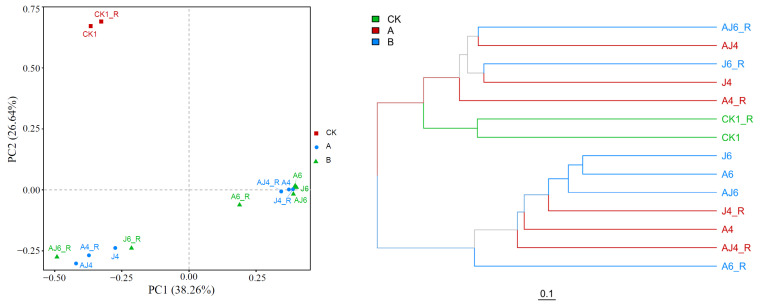
PCA diagram and clustering tree diagram under different processing conditions based on OUTs (group A represents each experimental group on the 56th day of the experiment, and group B represents each experimental group on the 84th day of the experiment).

**Figure 7 ijerph-20-02767-f007:**
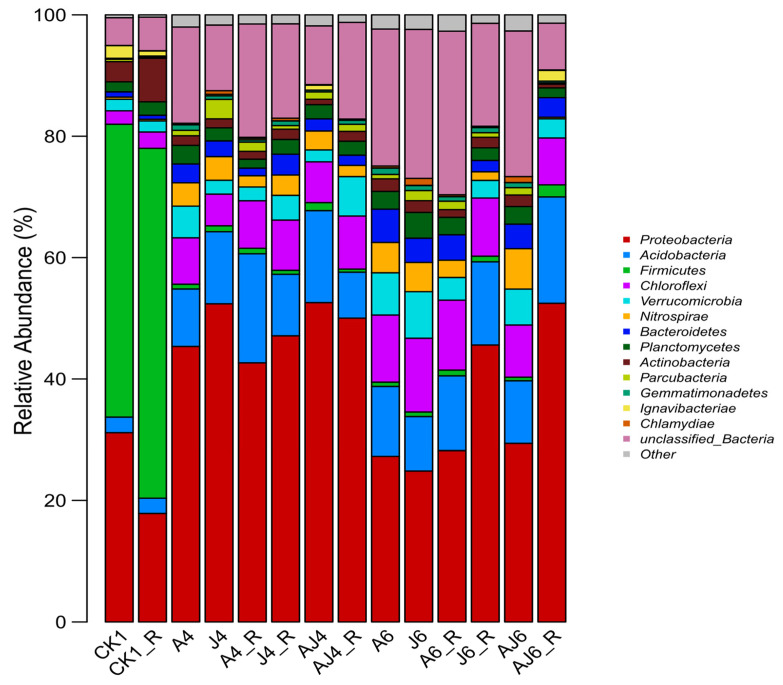
Relative abundance of microorganisms at the phylum level under different treatment conditions (only the top 10 species are shown, non-top-10 or non-detected species are indicated by Other).

**Figure 8 ijerph-20-02767-f008:**
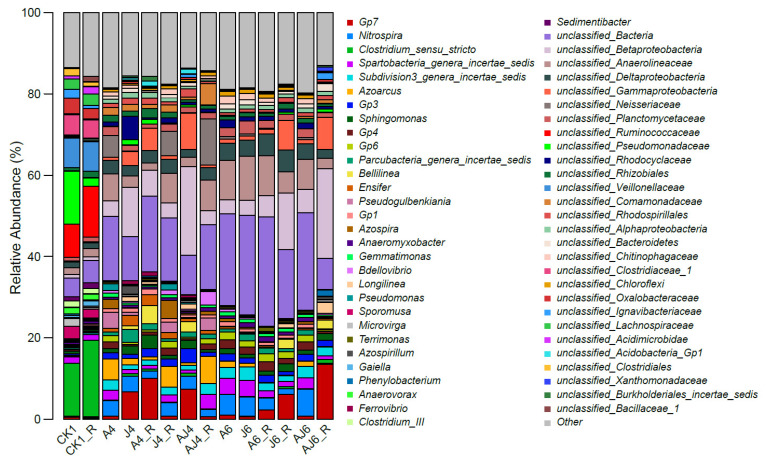
Relative abundance of bacteria at the genus level under different treatment conditions (only genera showing relative abundances greater than 1% during degradation of *β*-HCH; those at less than 1% and those not detected were classified as Other).

**Figure 9 ijerph-20-02767-f009:**
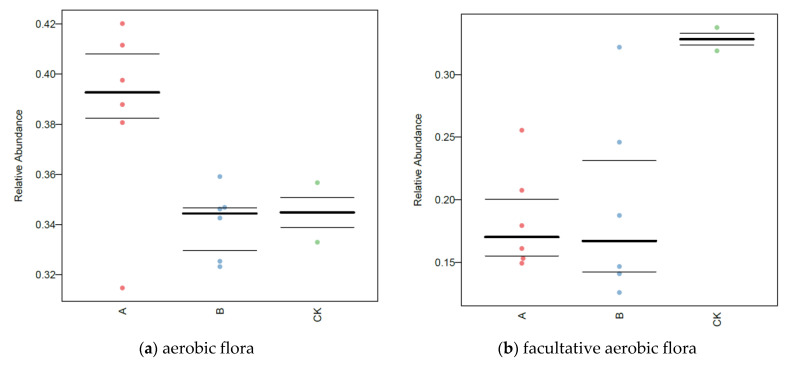
Phenotypes of microbiome samples based on BugBase analysis under different treatment conditions (horizontal axis is group name and vertical axis is relative abundance; group A represents each experimental group on the 56th day of the experiment and group B represents the 84th day of the experiment).

## Data Availability

Not applicable.

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
