# Peer review of "Degradation Characteristics and Remediation Ability of Contaminated Soils by Using β-HCH Degrading Bacteria"

_ijerph, 2023, doi:10.3390/ijerph20042767_

Round 1
Reviewer 1 Report
Manuscript entitled “Degradation characteristic and remediation ability of contaminated soils by β-HCH degradation bacteria” is written at quite good English level, with only few minor mistakes and not completely correct sentence construction, e.g.:
- The title suggests that bacteria are the main contamination of soils. I suggest it slight change to, e.g., “Degradation characteristics and remediation ability of contaminated soils by using β-HCH-degrading bacteria”.
- Lines 46-47: “Currently, many microorganisms with good degradation of β-HCH had been reported […].” –“with good degradation ability” sounds more natural.
- Lines 68-69: “The preparation of culture medium was sterilized at 121 °C for 20 min in an autoclave”. “Prepared”?
Below, please also find my minor comments and questions about the manuscript:
- The phylogenetic trees presented in Figure 1 are slightly too small and it is quite hard to read them. The same goes for Figures 6 and 8.
- Is it possible to add some comparison with literature to the chapters 3.3-3.5? I am especially interested how isolated strains fare in relation to other described ones, also used for degradation of β-HCH.
- There was no Supporting Information file included, while Authors mentioned the results presented there.
Concluding, I suggest minor revision of presented article, with possibility of acceptance after some changes.
Author Response
Comment 1: The title suggests that bacteria are the main contamination of soils. I suggest it slight change to, e.g., “Degradation characteristics and remediation ability of contaminated soils by using β-HCH-degrading bacteria”.
Response: Thanks for your suggestions. Manuscript entitled “Degradation characteristic and remediation ability of contaminated soils by β-HCH degradation bacteria” have changed to “Degradation characteristics and remediation ability of contaminated soils by using β-HCH-degrading bacteria”.
Comment 2: Lines 46-47: “Currently, many microorganisms with good degradation of β-HCH had been reported […].” –“with good degradation ability” sounds more natural.
Response: Thanks for your suggestions. In line 47, “…with good degradation…” have changed to “…with good degradation ability…”.
Comment 3: Lines 68-69: “The preparation of culture medium was sterilized at 121 °C for 20 min in an autoclave”. “Prepared”?
Response: Thanks for your suggestions. The culture medium is prepared after sterilization in an autoclave, and we also think it is more accurate to use prepared, so in line 68, changed “preparation” to “prepared”.
Comment 4: The phylogenetic trees presented in Figure 1 are slightly too small and it is quite hard to read them. The same goes for Figures 6 and 8.
Response: Thanks for your suggestions. We do apologize for the size of the legend and images, Figure 1, Figure 6 and Figure 8 have been revised to make them clearer.
Comment 5: Is it possible to add some comparison with literature to the chapters 3.3-3.5? I am especially interested how isolated strains fare in relation to other described ones, also used for degradation of β-HCH.
Response: Thanks for your suggestions. We added some comparison with literature to the chapters 3.3 in lines 202-210.
Comment 6: There was no Supporting Information file included, while Authors mentioned the results presented there.
Response: Thanks for your suggestions. We have supporting information that may have been uploaded or not provided in time and we will look on it.
Reviewer 2 Report
This work deals with study of bacteria for β-HCH degradation in soil. There are quite numerous characterizations, and the work is well-executed. The paper needs minor revisions before publication, here are some comments:
- Define acronyms the first time they appear.
- The goals of the work must be clearly stated at the end of the introduction.
- In Figure 4, it can be nice to add the degradation ability of the three strains alone for direct comparison purposes.
- A summary table with all the sample code names in the experimental section would be nice to have a global view of the different conditions and associated names.
Author Response
Comment 1: Define acronyms the first time they appear.
Response: Thanks for your suggestions. We have reviewed the manuscript throughout and found several acronyms that were not defined when they first appeared, which we have corrected. For example, the full name of β-HCH is given in line 11, and the full name of PCA is given in line 343.
Comment 2: The goals of the work must be clearly stated at the end of the introduction.
Response: Thanks for your suggestions. We added our goals at the end of the introduction in line 59-62: “The goals of this study is to find bacteria that have good degradation ability of β-HCH and can be used in the treatment of soil β-HCH contamination, even under low β-HCH concentration conditions, and provided a theoretical basis for the on-site engineering treatment of β-HCH contamination.”
Comment 3: In Figure 4, it can be nice to add the degradation ability of the three strains alone for direct comparison purposes.
Response: Thanks for your suggestions. We added the degradation ability of the three strains alone in Figure 4, and in line 280-283, we added some account: “It could be seen that only the degradation of β-HCH by the mixture of A1 and J1 was improved compared with that of the single bacteria, while the degradation of β-HCH by the other mixtures was generally reduced.”
Comment 4: A summary table with all the sample code names in the experimental section would be nice to have a global view of the different conditions and associated names.
Response: Thanks for your suggestions. About summary table, we have labelled and explained each step in the experimental section and adding another table would probably duplicate the previous narrative. However, if you think a summary table would be better, we can subsequently adapt the chapters and add a summary table.
Reviewer 3 Report
ijerph-2199377
Authors
Shape comments
In line 80, include the number and dimensions of wetlands used in the experiment
Comments
The research is well organized and structured
Author Response
Comment 1: In line 80, include the number and dimensions of wetlands used in the experiment
Response: Thanks for your suggestions. We added the number and dimensions of wetlands, also including their other usage parameters in line 83-87.